# Clinical and cognitive improvement following full-spectrum, high-cannabidiol treatment for anxiety: open-label data from a two-stage, phase 2 clinical trial

Mary Kathryn Dahlgren [1,2,3], Ashley M. Lambros[1,2], Rosemary T. Smith[1,2], Kelly A. Sagar[1,2,3], Celine El-Abboud[1,2] & Staci A. Gruber [1,2,3 ✉]

## Abstract

**Background** Evidence suggests cannabidiol (CBD) has anxiolytic properties, indicating potential for novel treatment strategies. However, few clinical trials of CBD-based products have been conducted, and none thus far have examined the impact of these products on cognition.

**Methods** For the open-label stage of clinical trial NCT02548559, autoregressive linear modeling assessed efficacy and tolerability of four-weeks of 1 mL t.i.d. treatment with a full-spectrum, high-CBD sublingual solution (9.97 mg/mL CBD, 0.23 mg/mL Δ−9-tetra-hydrocannabinol) in 14 outpatients with moderate-to-severe anxiety, defined as ≥16 on the Beck Anxiety Inventory (BAI) or ≥11 on the Overall Anxiety Severity and Impairment Scale (OASIS).

**Results** Findings suggest significant improvement on primary outcomes measuring anxiety and secondary outcomes assessing mood, sleep, quality of life, and cognition (specifically executive function) following treatment. Anxiety is significantly reduced at week 4 relative to baseline (BAI: 95% CI = [−21.03, −11.40], $p < 0.001$, OASIS: 95% CI = [−9.79, −6.07], $p < 0.001$). Clinically significant treatment response (≥15% symptom reduction) is achieved and maintained as early as week 1 in most patients (BAI = 78.6%, OASIS = 92.7%); cumulative frequency of treatment responders reached 100% by week 3. The study drug is well-tolerated, with high adherence/patient retention and no reported intoxication or serious adverse events. Minor side effects, including sleepiness/fatigue, increased energy, and dry mouth are infrequently endorsed.

**Conclusions** Results provide preliminary evidence supporting efficacy and tolerability of a full-spectrum, high-CBD product for anxiety. Patients quickly achieve and maintain symptom reduction with few side effects. A definitive assessment of the impact of this novel treatment on clinical symptoms and cognition will be ascertained in the ongoing double-blind, placebo-controlled stage.

### Plain language summary

Cannabidiol (CBD) is a compound found within the Cannabis sativa plant. Previous studies suggest CBD may reduce anxiety. In this clinical trial, 14 patients with anxiety were treated for four-weeks with a cannabis-derived study product with high levels of CBD, administered under their tongue 3 times each day. All patients knew that they were being given CBD. Following four weeks of treatment, patients reported reduced anxiety as well as improvements in mood, sleep, quality of life, and measures reflecting their self-control and ability to think flexibly. Patients did not experience any serious negative effects during the trial. The impact of this product is now being evaluated in more patients with anxiety.

[1] Cognitive and Clinical Neuroimaging Core, McLean Hospital Imaging Center, 115 Mill Street, Belmont, MA 02478, USA. [2] Marijuana Investigations for Neuroscientific Discovery (MIND) Program, McLean Hospital Imaging Center, 115 Mill Street, Belmont, MA 02478, USA. [3] Department of Psychiatry, Harvard Medical School, 25 Shattuck Street, Boston, MA 02115, USA. ✉email: gruber@mclean.harvard.edu

Approximately 34% of adults in the US are diagnosed with an anxiety disorder at some point in their lifetime[1]. A range of conventional pharmacotherapeutic agents are available, with some patients achieving adequate clinical response. While efficacious for many, conventional medications are typically associated with delayed symptom relief, with full treatment response taking up to 12 weeks[2]. Additionally, more than half of patients report bothersome side effects including cognitive and sexual dysfunction[3]. Importantly, current treatment options do not guarantee adequate symptom alleviation. Many patients do not achieve remission status[4], and approximately 41% of patients with anxiety go untreated[5], indicating a critical clinical need for efficacious, alternative or adjunctive treatment options with fewer side effects.

Extensive preclinical research has demonstrated the anxiolytic effects of cannabidiol (CBD), a primary non-intoxicating constituent of cannabis, in several animal models[6]; however, relatively few human studies have investigated the anxiolytic effects of CBD[7]. Acute administration studies have demonstrated anxiolytic effects of CBD in healthy adults during pharmacologically-induced anxiety[8] and a simulated public speaking task (SPST)[9] as well as in patients with social anxiety disorder (SAD)[10], including during the SPST[11]. To date, only one double-blind clinical trial has examined CBD for anxiety; teenagers with SAD exhibited reduced anxiety following 4 weeks of CBD treatment (300 mg/day)[12]. Interestingly, these investigations all used study products containing single extracted compounds (CBD only), which exhibit bell-shaped dose-response curves with limited dosage range for therapeutic response[13]. However, preliminary research suggests that whole-plant, full-spectrum products with diverse cannabinoid profiles may yield therapeutic response at lower doses relative to single extracted compounds, potentially due to the synergistic effects of multiple cannabinoids and the presence of other compounds (e.g., terpenoids, flavonoids)[14,15].

Additionally, previous investigations have primarily focused on clinical symptoms with few examining cognition, and none have used well-validated, robust cognitive assessments. Although some studies utilizing self-report scales noted cognition-related improvements (e.g., clear-headedness, mental sedation) following acute administration of CBD[8,11], cognitive outcomes were not assessed in the only previous clinical trial examining the impact of CBD in patients with SAD[12]. However, some observational studies assessing medical cannabis patients, who often report frequent use of high-CBD products, have demonstrated significant improvements on objective, clinician-administered neuropsychological assessments following initiation of treatment, particularly on measures of executive function[16–18], suggesting medical cannabis treatment may not result in negative cognitive outcomes. However, given chronic, heavy, recreational cannabis use is associated with cognitive decrements across a range of domains, including executive functioning and memory[19], understanding the impact of medical cannabis treatment (including the use of high-CBD products) on cognition is particularly important, and additional research is needed using well-validated cognitive assessments.

The current investigation reflects data from the open-label stage of clinical trial NCT02548559, designed to examine 4 weeks of treatment with a whole-plant, full-spectrum, high-CBD sublingual product in patients with anxiety. As previous research has only examined the impact of purified, single compound CBD study products and not full-spectrum products, the open-label stage of this trial was designed to gather safety and efficacy data on the novel full-spectrum study product to help inform dosing for the double-blind stage. Since extensive preclinical and human studies have demonstrated reduced anxiety using CBD, our primary outcome hypothesis was that patients would exhibit reduced anxiety following treatment relative to baseline. We also hypothesized that patients would demonstrate improvement on secondary outcomes related to clinical state (e.g., mood, sleep, quality of life). Additionally, cognitive performance was also assessed as a secondary outcome; given findings from observational studies, we hypothesized that patients would demonstrate improved executive function following treatment with the study product.

## Methods

**Study design and participants**. This study was approved by the Mass General Brigham Institutional Review Board (IRB) and carried out in accordance with the Declaration of Helsinki. Patients provided written informed consent to voluntarily participate in study procedures. As stipulated in our IRB-approved protocol, analyses from the open-label stage were required before initiating the double-blind stage of the study in order to ensure clinical efficacy at the selected dose. Study enrollment was completed at McLean Hospital between June 2018–February 2020. The study timeline included a baseline visit with weekly check-in visits during the 4-week trial (5 visits total). Participants were compensated $350 for completing the study.

Outpatients reporting at least moderate levels of anxiety, defined as ≥16 on the Beck Anxiety Inventory (BAI)[20] or ≥11 on the Overall Anxiety Severity and Impairment Scale (OASIS)[21] at baseline, were recruited from the New England area through online advertisements and social media postings. Patients were required to have a stable pharmacotherapeutic regimen with no changes in the past three months. History of previous cannabis and cannabinoid use was assessed using a modified version of the Timeline Followback[22], which queried both recreational and medical cannabis use. For this assessment, we specified that the term cannabis referred to marijuana, hemp, CBD, or any cannabinoid-containing product. To minimize effects of previous cannabinoid exposure, patients were required to be either cannabis naïve (<15 lifetime uses) or abstinent from regular use (defined as >1x/month) for ≥1 year prior to their baseline visit. Patients provided urine samples at each visit which were assessed using CLIA-waived drug assays. At baseline, patients were required to test negative for metabolites of $\Delta$−9-tetrahydrocannabinol (THC), the primary intoxicating constituent in cannabis. At follow-up visits, positive urine samples were sent to an outside laboratory for quantification of THC metabolites (results already published[23]). Additionally, patients were asked about their substance use at weekly check-in visits throughout the 4-week trial. Patients were disqualified from the trial if they endorsed use of any cannabis/cannabinoid-based products other than the study product. Full inclusion/exclusion criteria are provided in Fig. 1. The open-label stage was initially designed to assess 16 patients in order to assess efficacy, safety, and dosing for the double-blind stage; however, the onset of the COVID-19 pandemic and subsequent prohibition of in-person visits for research studies resulted in early termination of the open-label stage. At the time enrollment closed for the open-label arm, 15 patients had already been enrolled and complete data was available for 14 patients.

The study product was formulated from a full-spectrum, high-CBD base extract from cannabis provided by the National Institute on Drug Abuse. The CBD extract was homogenized into a solution containing medium chain triglyceride oil with an emulsifier, polysorbate-80, and dispensed into 30 mL glass bottles. The final study product contained 9.97 mg/mL CBD (1.04%) and 0.23 mg/mL THC (0.02%), confirmed by ProVerde Laboratories

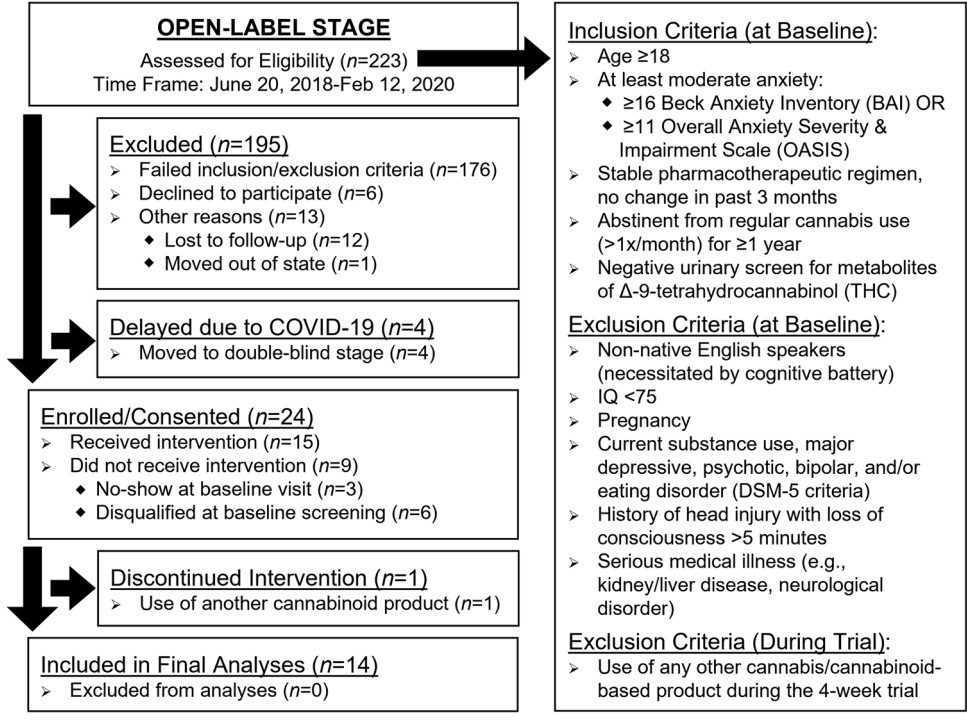

**Fig. 1 Recruitment and enrollment.** CONSORT flow chart of recruitment and enrollment (including inclusion/exclusion criteria) for the open-label stage of clinical trial NCT02548559 examining a high-cannabidiol (CBD) sublingual product for anxiety.

(Milford, MA). Information about other cannabinoids is presented in Table 1. For this 4-week trial, patients were instructed to use the calibrated dropper to self-administer 1 mL of the study product sublingually and hold for a minimum of 60 s, 3 times per day. The targeted daily dose was 30 mg CBD and <1 mg THC. Actual product use was quantified via outgoing versus incoming bottle weights cross-referenced with drug diaries, which were reviewed at weekly check-in visits.

**Clinical scales**. At all visits, patients completed a check-in with study staff and comprehensive self-report and clinician-rated assessments measuring the primary outcome of anxiety. The BAI and OASIS are brief self-report measures of general anxiety and anxiety-related impairment and were used to determine study eligibility. The State-Trait Anxiety Inventory (STAI)[24] is a self-report scale that differentiates between the temporary condition of state (how patients *currently* feel) and the long-lasting quality of trait (how patients *generally* feel). Lastly, the Hamilton Anxiety Rating Scale (HAM-A)[25] is a clinician-rated scale, and therefore less likely biased by patient self-report. After 4 weeks of treatment, patients rated their perception of change using a Patient Global Impression of Change (PGIC)[26] scale ranked from 1, "no change or condition has gotten worse," to 7, "a great deal better and a considerable improvement that has made all the difference."

Patients also completed self-report assessments measuring secondary clinical outcomes of mood and depressive symptoms, sleep disturbance, sexual function, and quality of life. The Beck Depression Inventory (BDI)[27], Profile of Mood States Total Mood Disturbance (POMS TMD)[28], and the Positive and Negative Affect Schedule (PANAS)[29] were completed at each visit. Additional scales were completed at baseline and week 4, including the Beck Hopelessness Scale (BHS)[30], Beck Scale for Suicide Ideation (BSS)[31], Pittsburgh Sleep Quality Index (PSQI)[32], Arizona Sexual Experience Scale (ASEX)[33], and the Medical Outcomes Survey Short Form-36 (SF-36; domains: physical functioning, physical role limitations, emotional role limitations, energy/fatigue, emotional well-being, social functioning, pain, and general health)[34].

**Cognitive assessments**. At baseline and week 4, patients completed a comprehensive cognitive battery designed to assess secondary outcomes of cognition; this battery assessed executive function and memory[35], domains shown to be vulnerable to recreational cannabis use. Executive function was assessed using the Stroop Color Word Test, Trail Making Test (TMT), computerized Wisconsin Card Sorting Task (WCST)[36], Multi-Source Interference Task (MSIT)[37], Letter-Number Sequencing (LNS) subtest of the Wechsler Adult Intelligence Scale-Revised, Digit Symbol Substitution Task (DSST), and the Controlled Oral Word Association Test (COWAT). Visual memory was assessed using the Benton Visual Retention Task (BVRT) and verbal memory via the Rey Auditory Verbal Learning Task (RAVLT). To limit practice effects, alternate test forms were used at the follow-up assessment following 4 weeks of treatment for all tasks except Stroop, WCST, and MSIT.

**Statistics and reproducibility**. Data were collected and managed using REDCap[38,39] and analyzed using SPSS v.24. Linear mixed model analyses (SPSS syntax: MIXED) with first-order autoregressive AR(1) covariance structures (reference group = baseline) were conducted to assess mean change of outcome variables from baseline. The BAI, OASIS, HAM-A, STAI, BDI, POMS, and PANAS scores were assessed across five timepoints (baseline and weeks 1–4 of treatment) while BHS, BSS, PSQI, ASEX, SF-36, and all cognitive assessments were assessed across two timepoints (baseline and week 4). All analyses were two-tailed and corrected for multiple comparisons based on number of timepoints assessed (i.e., $\alpha = 0.05/5 = 0.010$ and $\alpha = 0.05/2 = 0.025$). To assess the impact of positive and negative expectancy effects, all patients completed the Marijuana Effect Expectancy Questionnaire-Brief (MEEQ-B)[40] at baseline; Pearson's correlations assessed the association between positive and negative expectancies

**Table 1 Demographic, study product use, and patient impression of change analyses for patients who completed the open-label stage of a high-cannabidiol (CBD) study product for anxiety.**

| Demographics | Patients (n = 14) | | |
|---|---|---|---|
| | n (%) or Mean ± SD | Minimum | Maximum |
| Gender identity | | | |
| Female | 11 (78.6%) | - | - |
| Male | 3 (21.4%) | - | - |
| Age | 41.36 ± 16.89 | 22 | 64 |
| Estimated IQ (WASI) | 119.36 ± 5.51 | 106 | 129 |
| Body mass index (BMI) | 25.93 ± 4.07 | 20.52 | 34.54 |
| Race | | | |
| White | 12 (85.7%) | - | - |
| Black | 2 (14.3%) | - | - |
| Past cannabis use | | | |
| Cannabis naïve[a] | 5 (35.7%) | - | - |
| Past light use[b] | 6 (42.9%) | - | - |
| Past frequent use[c] | 3 (21.4%) | - | - |
| Abstinence from cannabis (yrs)[d] | 14.44 ± 13.65 | 3 | 37 |
| High-cannabidiol (CBD) study product use | | | |
| Treatment days | 31.07 ± 3.67 | 27 | 37 |
| Product use (mL/day) | 3.48 ± 0.60 | 2.32 | 4.70 |
| Exposure to specific cannabinoids[e] | | | |
| Cannabidiol (CBD; mg/day) | 34.73 ± 6.03 | 23.13 | 46.84 |
| Δ9-Tetrahydrocannabinol (THC; mg/day) | 0.80 ± 0.14 | 0.53 | 1.08 |
| Cannabichromene (CBC; mg/day) | 0.97 ± 0.17 | 0.65 | 1.32 |
| Cannabigerol (CBG; mg/day) | 0.38 ± 0.07 | 0.26 | 0.52 |
| Cannabinol (CBN; mg/day) | 0.21 ± 0.04 | 0.14 | 0.28 |
| Patient global impression of change (PGIC) | Median (IQR) | | |
| PGIC after 4 weeks of treatment | 6 (1.25) | 4 | 6 |

Abbreviations: *IQR* interquartile range, *WASI* Wechsler Abbreviated Scale of Intelligence.
[a]Cannabis naïve at baseline was defined as ≤15 lifetime uses & <1 use/month.
[b]Past light cannabis use was defined as a previous period of ≥1 use/month & ≤2 uses/week.
[c]Past frequent cannabis use was defined as a previous period of ≥3 uses/week.
[d]Only patients with a history of past cannabis use (n = 9) reported years of abstinence.
[e]The study product was also assessed for cannabidiolic acid (CBDa), cannabidivarin (CBDV), cannabigerolic acid (CBGa), tetrahydrocannabinolic acid (THCa), and tetrahydrocannabivarin (THCV), which were not present above the limit of detection.

and the difference score (baseline minus week 4) of any clinical variable that changed significantly following treatment. Percent change relative to baseline was calculated for BAI and OASIS scores for all follow-up assessments to generate cumulative frequencies of treatment responders, defined as patients who achieved and maintained clinical improvement of ≥15% reduction in anxiety scores on either the BAI or OASIS. This threshold was selected based on previous work demonstrating that ≥15% symptom reduction was optimal for identifying those who had responded to treatment[41,42].

**Reporting summary**. Further information on research design is available in the Nature Research Reporting Summary linked to this article.

## Results

**Demographics**. Fourteen outpatients (11 females, 3 males) aged 41.36 ± 16.89 completed the entire trial (Table 1). Of enrolled patients who received study product (n = 15), one was discontinued and excluded from analyses for use of another cannabinoid product during the trial. Only minimal data were missing from analyses and were considered missing completely at random (e.g., lab error processing urine sample).

**Study product use**. Patients used an average of 3.48 ± 0.60 mL/day of study product, equivalent to 34.73 ± 6.03 mg/day CBD and 0.80 ± 0.14 mg/day THC. The study drug was well-tolerated with no reported intoxication, serious adverse events, or major protocol deviations. Some minor side effects were noted; sleepiness/fatigue, increased energy, and dry mouth were the most

frequently endorsed (ns = 3, Table 2). At week 4, side effect severity was predominantly ranked as mild except for three patients who each ranked one symptom as moderate (increased energy, increased appetite, or acid reflux). Throughout the trial, no side effects were rated as severe.

**Clinical state and treatment response**. Following 4 weeks of treatment, significant decreases were noted on all primary outcome assessments of anxiety relative to baseline (Table 3) on self-report measures, including the BAI (95% CI −21.03, −11.40; Fig. 2A), OASIS (95% CI −9.79, −6.07; Fig. 2B), STAI state (95% CI −19.83, −6.32; Fig. 2D), and STAI trait (95% CI −21.00, −10.57; Fig. 2E), as well as the clinician-rated HAM-A (95% CI −19.25, −13.60; Fig. 2F; source data in Supplementary Data). Significant treatment response was observed as early as week 1 with most patients achieving and maintaining ≥15% reduction of BAI (78.6%) and OASIS (92.7%) scores at this timepoint; the cumulative frequency of treatment responders was 100% by week 3 of treatment (Fig. 2C). Further, following 4 weeks of treatment, the median rating of patients' impression of change was 6 (IQR = 1.25), "better and a definite improvement that has made a real and worthwhile difference." Secondary outcome assessments of mood, sleep, and quality of life demonstrated significant improvement following 4 weeks of treatment (Supplementary Table 1), with reduced symptoms of depression and negative affect, improved mood and sleep, and increased quality of life. Ratings of sexual dysfunction remained stable over time. Additionally, neither positive nor negative expectancy effects measured

**Table 2 Frequency of side effects reported at treatment week 4 with a high-cannabidiol (CBD) study product for anxiety.**

| Side Effects | Patients (n = 14) n (%) |
|---|---|
| Sleep | |
| Sleepiness/fatigue | 3 (21.4%) |
| Sleeping more | 2 (14.3%) |
| Sleeping less | 1 (7.1%) |
| Energy | |
| Increased energy* | 3 (21.4%) |
| More talkative | 2 (14.3%) |
| Less talkative | 1 (7.1%) |
| Physiologic | |
| Dry mouth | 3 (21.4%) |
| Cognitive | |
| Cognitive cloudiness | 2 (14.3%) |
| Memory problems | 1 (7.1%) |
| Difficulty concentrating | 1 (7.1%) |
| Gastrointestinal | |
| Decreased appetite | 1 (7.1%) |
| Increased appetite* | 1 (7.1%) |
| Weight gain | 1 (7.1%) |
| Constipation | 1 (7.1%) |
| Anxiety | |
| Anxiety | 1 (7.1%) |
| Alcohol use | |
| Decreased alcohol use | 1 (7.1%) |
| Libido | |
| Increased libido | 1 (7.1%) |
| Other (write-in) | |
| Acid reflux* | 1 (7.1%) |

Unless otherwise noted, all side effects were rated as mild; no side effects were rated as severe.
*One person rated this side effect as moderate.
The following side effects were also queried, but were not endorsed by any patient:
Energy: lassitude, decreased energy, couch lock.
Physiologic: muscle twitching, dizziness, itchy skin, sweating, headache, urinary retention
Cognitive: feeling altered.
Gastrointestinal: nausea, abdominal discomfort, diarrhea, weight loss.
Anxiety: panic.
Alcohol Use: increased alcohol use.
Libido: decreased libido.
Cardiovascular/Respiratory: heart racing/palpitations, respiratory symptoms.
Mood/Psychiatric: apathy, depression, mania, racing thoughts, paranoia.

by MEEQ-B were associated with clinical improvements (Supplementary Table 2).

**Cognitive assessments**. Following 4 weeks of treatment, patients demonstrated significantly improved performance on secondary outcome assessments of cognition. Specifically, improved executive function characterized by faster completion time on the Interference condition of the Stroop Color Word Test (95% CI −17.66, −5.05); faster response time (95% CI −109.23, −23.82) and reduced commission errors (95% CI −7.12, −1.06) on the Interference condition of the MSIT; and increased concept level responses (95% CI 2.53, 11.61), total correct (95% CI 0.70, 6.87), and total categories (95% CI 0.16, 1.13) on the WCST were observed following treatment (Table 4). Performance remained stable on all other measures of executive function as well as on measures of visual and verbal memory.

## Discussion

Results from the open-label stage of this clinical trial provide preliminary evidence that 4 weeks of treatment with a full-spectrum, high-CBD sublingual product is efficacious in patients with moderate-to-severe anxiety, confirming and extending previous preclinical and clinical research[6,12]. Significant reductions for

**Table 3 Changes in anxiety ratings over the course of 4 weeks of treatment with a high-cannabidiol (CBD) sublingual study product: autoregressive linear mixed models (Two-Tailed).**

| Clinical Scale | Mixed Model F p (η_lp^2) | Baseline n = 14 (ref.) Mean [95% CI] | Week 1 n = 14 Estimate (p) [95% CI] | Week 2 n = 14 Estimate (p) [95% CI] | Week 3 n = 14 Estimate (p) [95% CI] | Week 4 n = 14 Estimate (p) [95% CI] | Baseline to Week 4 Percent Change |
|---|---|---|---|---|---|---|---|
| Anxiety scales | | | | | | | |
| Beck anxiety inventory (BAI) | **17.48** **<0.001 (0.65)** | 20.29 [16.31, 24.26] | **−11.07 (<0.001)** [−14.13, −8.02] | **−13.00 (<0.001)** [−16.96, −9.04] | **−14.86 (<0.001)** [−19.33, −10.38] | **−16.21 (<0.001)** [−21.03, −11.40] | −79.93% |
| Overall anxiety severity and impairment scale (OASIS) | **24.80** **<0.001 (0.75)** | 11.29 [9.93, 12.65] | **−5.57 (<0.001)** [−6.95, −4.19] | **−6.21 (<0.001)** [−7.89, −4.54] | **−7.00 (<0.001)** [−8.80, −5.20] | **−7.93 (<0.001)** [−9.79, −6.07] | −70.25% |
| Hamilton anxiety rating scale (HAM-A) | **58.12** **<0.001 (.84)** | 19.86 [17.68, 22.04] | **−13.00 (<0.001)** [−14.92, −11.08] | **−13.64 (<0.001)** [−16.06, −11.23] | **−15.64 (<0.001)** [−18.32, −12.97] | **−16.43 (<0.001)** [−19.25, −13.60] | −82.73% |
| State-trait anxiety inventory (STAI): state | **4.97** **0.002 (0.38)** | 46.00 [41.02, 50.98] | **−6.71 (0.009)** [−11.64, −1.79] | **−11.50 (<0.001)** [−17.51, −5.49] | **−9.71 (0.004)** [−16.22, −3.21] | **−13.07 (<0.001)** [−19.83, −6.32] | −28.42% |
| State-trait anxiety inventory (STAI): trait | **10.65** **<0.001 (0.67)** | 56.29 [51.78, 60.79] | **−7.29 (<0.001)** [−10.48, −4.09] | **−11.36 (<0.001)** [−15.55, −7.16] | **−13.50 (<0.001)** [−18.30, −8.70] | **−15.79 (<0.001)** [−21.00, −10.57] | −28.05% |

Bold numbers are significant at p ≤ 0.010 for 5 timepoints.
Significance is only noted for estimates relative to the baseline reference group.
Note: For the BAI, OASIS, HAM-A, and STAI, reduced estimates indicate clinical improvement.

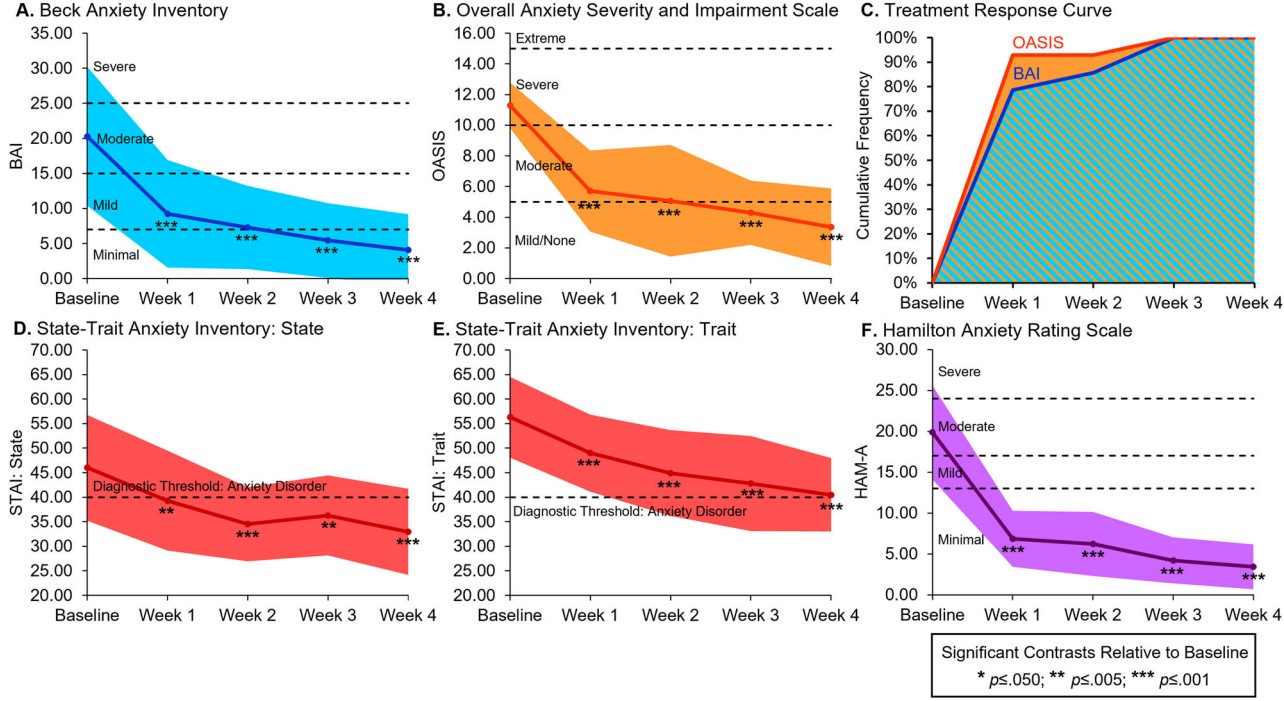

**Fig. 2 Line graphs of reduced anxiety ratings and cumulative frequency of treatment responders.** Line graphs demonstrating significant reductions in anxiety ratings in 14 patients following 4 weeks of treatment with a high-cannabidiol (CBD) study product for anxiety. Significant reductions in anxiety relative to baseline were observed for the (**A**) Beck Anxiety Inventory (BAI), (**B**) Overall Anxiety Severity and Impairment Scale (OASIS), (**D**) and (**E**) State-Trait Anxiety Inventory (STAI), and (**F**) Hamilton Anxiety Rating Scale (HAM-A). Additionally, cumulative frequency curves of treatment responders, defined as patients who achieved and maintained clinical improvement of ≥15% reduction of anxiety symptoms on the BAI and OASIS, demonstrated rapid, clinically significant treatment response (**C**). Most patients achieved and maintained treatment response after 1 week and all patients achieved and maintained treatment response by week 3. The color-shaded areas represent one standard deviation from the mean (**A**, **B**, **D**, **E**, **F**) or the proportion of treatment responders (**C**).

primary outcome measures of anxiety were detected across several clinical assessments including self-report scales, clinician-rated scales, and patient impression of change. At baseline, patients' average anxiety ratings were considered moderate on the BAI and severe on the OASIS, but following treatment, both BAI and OASIS average scores fell in the minimal or mild-to-none range of severity. Treatment response analyses revealed rapid onset of clinically significant reductions in anxiety with most patients achieving and maintaining treatment response after 1 week and all patients achieving and maintaining treatment response by week 3. This rapid response has been observed in previous clinical trials of cannabinoid-based products[12], and is a marked improvement over the typical time course (up to 12 weeks) often required for full treatment response using conventional pharmacotherapy[2].

Interestingly, in the current trial, treatment response was observed at a much lower dosage (~30 mg/day) than a previous trial using a single extracted CBD isolate (~300 mg/day)[12]. This difference may be related to the entourage effect, a term used to describe the potentially enhanced effects of cannabinoids when a variety of metabolites and closely related compounds (e.g., cannabinoids, terpenoids, flavonoids) work together synergistically[15]. While few studies have directly compared full-spectrum and single extracted products, research suggests that for some conditions, full-spectrum products may yield therapeutic response at lower doses and with fewer side effects. Specifically, a meta-analysis by Pamplona and colleagues[14] demonstrated that patients with refractory epilepsy treated with full-spectrum, high-CBD products reported lower average dose relative to those treated with single extracted CBD isolate products. Further, single extracted CBD products were associated with more frequent

reporting of mild and severe side effects relative to full-spectrum, high-CBD products. Additionally, preclinical research from Gallily and colleagues[13] reported a bell-shaped dose-response curve for the anti-inflammatory and anti-nociceptive effects of a single extracted CBD isolate, but a linear dose-response for a full-spectrum high-CBD product (17.9% CBD, 1.1% THC, plus other cannabinoids). The primary goal of the current study was to gather safety and efficacy data on the novel, full-spectrum study product to help inform dosing for the double-blind stage; however, the double-blind, placebo-controlled stage of the trial also includes a matched CBD isolate treatment arm, which will allow for direct comparison of full-spectrum and single extracted products.

The study drug was well-tolerated with no serious adverse events and few side effects. Interestingly, several reported side effects were deemed beneficial in addressing anxiety-related issues (e.g., sleeping more). The tolerability of CBD is another benefit relative to conventional pharmacotherapy, which is often associated with burdensome side effects[2]. Further, other pharmacotherapies (e.g., benzodiazepines) are associated with high abuse liability, but recent evidence suggests that patients who use cannabinoid-based products for medical purposes actually exhibit few signs of problematic use[43]. Additional clinical trials should compare efficacy of response and side effect profile of CBD-based products to conventional pharmacotherapy.

On secondary outcome measures assessing cognitive function, patients exhibited improved or stable performance following treatment. Specifically, patients exhibited significantly faster performance with fewer errors on several tasks of executive function relative to baseline, while assessments of visual and verbal memory remained stable. These findings are consistent

**Table 4 Changes in cognitive performance following 4 weeks of treatment with a high-cannabidiol (CBD) sublingual product: autoregressive linear mixed models (Two-Tailed).**

| Cognitive Assessment | Mixed Model F p ($\eta_p^2$) | Baseline n = 14 (ref.) Mean [95% CI] | Week 4 n = 14 Estimate (p) [95% CI] | Baseline to Week 4 Percent Change |
|---|---|---|---|---|
| **Stroop color word test** | | | | |
| Interference time (s) | **15.13** **0.002 (0.54)** | 101.93 [89.21, 114.65] | **−11.36 (0.002)** **[−17.66, −5.05]** | −11.14% |
| Interference accuracy (%) | 0.02 0.900 (<0.01) | 98.79 [98.03, 99.54] | −0.07 (0.900) [−1.28, 1.14] | −0.07% |
| **Trail making test (TMT)** | | | | |
| Trails B time (s) | 0.01 0.932 (<0.01) | 59.07 [42.48, 75.66] | −0.64 (0.932) [−16.55, 15.26] | −1.09% |
| Trails B errors | 1.09 0.315 (0.08) | 0.29 [−0.24, 0.81] | 0.36 (0.315) [−0.38, 1.09] | 125.00% |
| **Multi-source interference task (MSIT)**[a] | | | | |
| Interference response time (ms) | **12.05** **0.006 (0.55)** | 919.16 [842.86, 995.46] | **−66.53 (0.006)** **[−109.23, −23.82]** | −7.24% |
| Interference commissions | **9.07** **0.013 (0.48)** | 7.55 [5.32, 9.77] | **−4.09 (0.013)** **[−7.12, −1.06]** | −54.22% |
| Interference omissions | 1.37 0.268 (0.12) | 3.82 [−0.29, 7.93] | −3.27 (0.268) [−9.50, 2.95] | −85.71% |
| **Wisconsin card sorting task (WCST)** | | | | |
| Concept level response | **11.31** **0.005 (0.47)** | 36.71 [27.62, 45.81] | **7.07 (0.005)** **[2.53, 11.61]** | **19.26%** |
| Total correct | **7.04** **0.020 (0.35)** | 43.57 [37.10, 50.05] | **3.79 (0.020)** **[0.70, 6.87]** | **8.69%** |
| Total categories | **8.16** **0.013 (0.39)** | 2.86 [1.95, 3.76] | **0.64 (0.013)** **[0.16, 1.13]** | **22.50%** |
| Total perseverative errors | 3.79 0.073 (0.23) | 11.57 [5.70, 17.44] | −2.00 (0.073) [−4.22, 0.22] | −17.28% |
| **Letter-number sequencing (LNS)** | | | | |
| LNS total | 3.83 0.072 (0.23) | 12.50 [11.25, 13.75] | 0.93 (0.072) [−0.10, 1.95] | 7.43% |
| **Digit symbol substitution task (DSST)** | | | | |
| DSST total | 0.60 0.451 (0.04) | 60.57 [54.00, 67.14] | 1.21 (0.451) [−2.16, 4.59] | 2.00% |
| **Controlled oral word association task (COWAT)** | | | | |
| Adjusted score | 0.05 0.836 (0.05) | 50.79 [42.54, 59.03] | 0.36 (0.836) [−3.29, 4.01] | 0.70% |
| Categories | 1.71 0.213 (0.12) | 25.43 [22.06, 28.80] | −1.93 (0.213) [−5.11, 1.25] | −7.58% |
| **Benton visual retention task (BVRT)** | | | | |
| BVRT total | 0.21 0.657 (0.02) | 7.50 [6.41, 8.59] | −0.21 (0.657) [−1.23, 0.80] | −2.86% |
| **Rey auditory verbal learning task (RAVLT)** | | | | |
| Trials 1–5: total correct | 3.74 0.075 (0.22) | 52.14 [47.65, 56.54] | −3.00 (0.075) [−6.35, 0.35] | −5.75% |
| Short delay: correct | 0.39 0.542 (0.03) | 10.50 [8.61, 12.39] | −0.36 (0.542) [−1.59, 0.88] | −3.40% |
| Long delay: correct | 0.07 0.800 (0.01) | 10.29 [8.41, 12.16] | −0.14 (0.800) [−1.34, 1.05] | −1.39% |

Bold numbers are significant at $p \leq 0.025$ for 2 timepoints.
Significance is only noted for estimates relative to the baseline reference group.
[a] n = 11: first 3 patients did not complete the MSIT.

with data from observational investigations reporting long-term treatment with medical cannabis is associated with improved clinical state and executive functioning[16–18] as well as acute CBD administration studies indicating significant cognitive improvement on self-report assessments[8,11]. Interestingly, these findings contrast with research on chronic, recreational cannabis use, which is typically associated with *poorer* cognitive performance. Importantly, however, differences in cognitive outcomes between recreational consumers and medical cannabis patients are likely related to differences in cannabis-related variables, such as age of onset and notable differences in exposure to specific cannabinoids (e.g., higher THC and very low to no discernible CBD content in

the majority of recreational consumer products)[19]. Lastly, extensive research indicates that anxiety impairs cognitive function, suggesting that patients' performance is likely to improve with reduction of clinical symptomatology[44]. Future studies should continue to assess the impact of CBD and other cannabinoids on cognition as well as the role of symptom alleviation.

**Limitations**. The current study presents data from the open-label stage of a clinical trial with the primary goal of determining dosing efficacy and tolerability in a small sample of patients. Open-label designs can be biased by treatment expectancies, as

both patients and research staff are unblinded regarding study product status. Importantly, in the current trial, positive and negative expectancies at baseline did not correlate with clinical improvement for patients. However, it is important to note that an expectancy measure for medical cannabis treatment (e.g., CBD) does not exist. Therefore, we utilized the MEEQ-B, a well-validated metric designed to assess expectancies related to recreational cannabis use (i.e., THC exposure)[40], and instructed patients to rate expectancies regarding the study product instead of general cannabis/marijuana use. The MEEQ-B assesses positive (e.g., feeling calm, reducing tension) as well as negative (e.g., feeling high, altered perception) expectancies related to cannabis use, which likely (albeit indirectly) impact treatment expectancies. The MEEQ-B was selected to ensure that any bias regarding cannabis use was assessed. In order to specifically assess medical cannabis treatment-related expectancies, we created a measure currently in use in the ongoing double-blind, placebo-controlled stage of this trial.

It is also important to recognize that regression to the mean is potentially problematic in clinical research, particularly when assessing patients with clinically significant symptomatology, as extreme values trend toward the population mean upon repeated sampling[45]. This can impact internal validity of these studies and reduce confidence in the causal link between the independent and dependent variables (e.g., treatment and clinical outcomes). Randomized, placebo-controlled, clinical trials can help differentiate between improvement related to treatment and improvement related to regression to the mean, as regression to the mean has been hypothesized to markedly contribute to placebo effects[45]. The ongoing placebo-controlled stage of this trial will further address these issues. It is of note that in the current study, the large effect sizes observed for the primary clinical outcomes measuring anxiety suggest results are not wholly attributable to regression to the mean. Further, baseline values of cognitive assessments were not extreme, suggesting that the observed improvements of executive functioning are also not likely solely based on regression to the mean.

In the current study, patients were primarily White women with above average IQ, potentially limiting the generalizability of results. Epidemiological studies indicate that White Americans are significantly more likely to be diagnosed with general anxiety disorder[46], but evidence suggests greater persistence (≥12 months) of mental health disorders among non-White minorities, with lower educational attainment and birthplace (i.e., US-born) associated with greater persistence of mental health disorders[47]. Additionally, lifetime prevalence statistics indicate that women are ~1.5 times more likely to have an anxiety disorder than men[1], whereas the gender distribution of our sample (78.6% female) included slightly higher numbers of women relative to population prevalence. Research on sex differences associated with CBD is limited, although evidence from preclinical and acute administration studies suggest that sex differences (and sex*age interactions) significantly impact the anxiolytic effects of THC[48,49]. Further, cytochrome P450 (CYP) enzymes, which are responsible for many metabolic processes including drug metabolism and clearance, are significantly impacted by sex, age, and ethnicity[50], which may impact the metabolism of CBD and other cannabinoids. Future studies should confirm efficacy of CBD-containing products for anxiety in underserved and underrepresented patient samples as well as comprehensively assess potential sex-specific effects of CBD.

Additionally, restrictions on in-person research due to the onset of the COVID-19 pandemic resulted in enrollment of a slightly smaller sample size than suggested by our a priori power analyses. However, power analyses for 5 repeated-measurements (at power = 0.90 and α = 0.05) indicated that shifting the sample size from 16 to 14 patient completers only slightly impacted the required effect size ($\eta^2 = 0.11$ vs $\eta^2 = 0.12$). Further, the lowest observed effect size for the primary assessments of anxiety in the current analyses was $\eta^2 = 0.38$ (for STAI state anxiety), more than double the required effect size from both power analyses. This suggests the study is well-powered to assess the primary outcome variables despite the slightly smaller final sample size than originally anticipated.

Lastly, while clinical trials are needed to evaluate risks and benefits of cannabinoid-based products, many patients already have access to a variety of cannabis products, underscoring the need for additional research on the efficacy and safely of products readily available to consumers. In particular, a number of cannabinoids, including CBD, interact with CYP enzymes, which may result in pharmacokinetic interactions with other medications and increase the chance of side effects[51,52]. Although no serious adverse effects were reported in this trial, future investigations should examine potential drug–drug interactions between conventional medications and individual cannabinoids.

## Conclusions

Initial results from the open-label stage of this clinical trial demonstrated significant improvement of primary outcome assessments of anxiety, providing preliminary evidence that a full-spectrum, high-CBD product may be efficacious for treating anxiety with few side effects. Patients quickly achieved and maintained ≥15% reduction of anxiety symptoms with most patients demonstrating clinically significant treatment response after 1 week of treatment. Treatment response was demonstrated at a much lower dosage than a previous clinical trial using a single extracted CBD isolate product. Secondary outcome assessments demonstrated improvements on measures of mood, sleep disturbance, quality of life and executive functioning following treatment. A definitive assessment of the impact of this novel treatment on clinical symptoms and cognition will be ascertained in the ongoing double-blind, placebo-controlled stage.

**Data availability**

The source data for the primary outcomes have been supplied as Supplementary Data. Additional datasets generated and/or analyzed during the open-label stage of the current study are available from the corresponding author upon reasonable request starting immediately after publication and ending 3 years after publication.

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

## Acknowledgements

The cannabis extract base for the study drug was provided by the National Institute on Drug Abuse (NIDA). Funding support for this project was generously provided by private donations to the Marijuana Investigations for Neuroscientific Discovery (MIND) program at McLean Hospital. This project was registered at clinicaltrials.gov (NCT02548559); additional project details (e.g., protocol, etc.) are provided online. Clinical consultation for this project was provided by David Olson, M.D., Ph.D. and Franca Centorrino, M.D.; these licensed physicians assisted with screening patients and co-signed for enrollment. Additionally, the authors would like to thank ProVerde Laboratories and Christopher Hudalla, Ph.D. for his role in providing laboratory services and consultation and Scott Lukas, Ph.D. for consultation regarding regulatory and compliance issues. Drs. Olson, Centorrino, Hudalla, and Lukas were not compensated for their contributions to this project.

## Author contributions

S.A.G. conceptualized and designed the study. All authors contributed to the acquisition of data (primarily S.A.G., A.M.L., & C.E.). M.K.D. performed all the statistical analyses and drafted the manuscript. R.T.S. provided administrative and regulatory support. M.K.D., S.A.G., & K.A.S. provided critical revision of the manuscript for important intellectual content.

## Competing interests

The authors declare the following competing interests: S.A.G. reported grants from the National Institute on Drug Abuse, Foria/Praxis Ventures, and Charlotte's Web outside the submitted work. Over the last five years, S.A.G. has reported receiving fees from the Coalition for Cannabis Policy, Education, and Regulation (CPEAR), Beth Israel Deaconess, Fenway Health, Greenwich Biosciences Cannabis Education Working Group, National Academy of Neuropsychology, McMaster University, Harvard Health Publications, and the Massachusetts College of Pharmacy and Health Sciences, all related to presentations outside the submitted work. S.A.G. is also a scientific advisor for Ajna Biosciences. M.K.D. reported receiving the McLean Hospital Jonathan Edward Brooking Mental Health Research Fellowship outside the submitted work. K.A.S. reported receiving the McLean Hospital Eleanor and Miles Shore Fellowship and the Charles Robert Broderick III Phytocannabinoid Research Fellowship, and reported receiving fees from CPEAR outside the submitted work. No other disclosures were reported. All other authors have no competing interests to declare. No funding sources were involved in the design and conduct of the study; collection, management, analysis, and interpretation of the data; preparation, review, or approval of the manuscript; and decision to submit the manuscript for publication.
