## [Peer Review File · Communications Medicine]

Reviewers' comments:

Reviewer #1 (Remarks to the Author):

This manuscript provides a detailed description of a well-conducted 4-week, open label pilot trial of “full-spectrum CBD” for 14 persons with elevated levels of anxiety who responded to online or social media ads to participate. Excellent assessment and outcome measures were used, and study methods resulted in excellent adherence to the study protocol with little missing data. Measures of anxiety and cognitive functioning improved substantially following 4 weeks of the study drug. Study findings are presented clearly to the reader. The findings will be of great interest to the scientific community and clinical medicine. The only primary concern with the manuscript is the overall tendency to use language and make statements that overreach what is known in the literature about CBD’s clinical effects and that overreach what can be concluded from a relatively small open label trial with study volunteers. As the authors convey in the Discussion, these findings await confirmation in a rigorously controlled study. A more conservative write up might be considered to not inadvertently mislead the readership, particularly given the political and cultural divides related to cannabis and cannabinoid compounds. Below I offer some comments and suggestions mostly related to this concern.

Perhaps consider use of less superlative adjectives throughout related to promise, potential, and outcomes of CBD, e.g., the use of “great” in first line of the abstract to modify potential.

Perhaps add adherence/retention data to the abstract if there is space

For this journal, full spectrum vs. single extracted compounds, as referred to in the abstract, may need more explanation if the readership is to glean its potential importance.

Consider reworking the first paragraph to position CBD as a potential alternative or adjunct medication rather than suggesting that other medications are highly flawed and problematic and that CBD could replace them (i.e., a panacea).

p. 5 line 89. It was unclear why language switched to “medical cannabis treatment” (rather than CBD) on recreational cannabis consumers, as referred to here, aren’t CBD users – this breeds unnecessary confusion for the reader (and eventually the public).

Perhaps the intro could better introduce the reader to what differentiates “whole-plant, full-spectrum, high CBD” from CBD only referred to earlier. i.e., what else is in the compound. Why is the product referred to as “novel”?

P5. Line 106 refers to but does not provide a cite(s) for “observational studies” that show CBD improved EF.

METHOD

Is it possible to list the other cannabinoids in the CBD product used in this study other than CBD and D9-THC?

Perhaps provide the rationale for considering a 15% reduction in anxiety scores “clinical improvement” – this could reflect a change from 20 to 17, which would be difficult to justify as clinically meaningful, particularly in an open label study or only 4 weeks. I do recognize that the results appear to show consistently larger reduction than 15%.

Given the cognitive data feature prominently in the purpose of the study, perhaps the data and analyses might better be placed in the primary report, not supplemental analyses.

The issue of power raised in the methods, then again in the discussion, does not seem highly relevant to a small open label trial – particularly given all participants responded in one direction and produced highly significance results (consider omitting?). This comes across as a little defensive, unnecessarily, and adds to the trend of arguing back against basic study weaknesses that are inherent to an open label trial.

Discussion

The discussion mentions all the limitations and weaknesses of the study, primarily related to what are always the weaknesses with an open label test of a medication. I find that the Discussion oversells the findings given these weaknesses – unnecessarily. More conservative language related to what was observed and how one might interpret the findings given the study design might be prudent. Also a more conservative discussion of the limitations without the subsequent text trying to discount the limitations would provide the reader with a more balanced presentation and interpretation of what was learned in this open label. Concluding efficacy before running and obtaining results from a rigorously controlled study is always problematic – and unfortunately in clinical science most positive findings are not supported by future trials. We can hope that will not be the case with this cannabinoid compound, but we must wait for the future results.

Reviewer #2 (Remarks to the Author):

This open label study aimed to assess the effects of a low dose full spectrum high CBD sublingual administration on anxiety and cognitive function among participants with elevated anxiety. Significant decreases in anxiety were observed at all four timepoints compared to baseline. Additionally, improvements in cognitive functioning were observed across various endpoints. These findings add to the small but growing empirical literature regarding the anxiolytic effects of CBD.

The detail in terms of methodology was appropriate for the aim of reproducibility, and the statistical approach was adequate. Below are suggestions aimed at improving the manuscript.

1. In the consort chart, please clarify reasons for exclusion as part of the Figure caption. Relatedly, it is unconventional to have two indices for the primary (anxiety) inclusion criterion (i.e., minimum of 16 on the Beck Anxiety Inventory [BAI] OR a minimum of 11 on the Overall Anxiety Severity and Impairment Scale [OASIS]). Provide additional detail regarding the rationale for this approach.
2. Please include baseline correlations among the primary outcome variables in the supplemental

material. Although the authors make a conceptual argument that these are distinct, the empirical overlap is likely quite large and may point to the need for further corrections.

3. Expectancy effects are a potent threat to the current findings. The authors argue that expectancies measured at baseline do not relate to clinical outcomes, increasing confidence in the observed findings. That is a reasonable effort to empirically address expectancy, but it is not super compelling in the context of the current study. For example, because a CBD expectancy measure does not currently exist, the authors adapted the Marijuana Effect Expectancy Questionnaire-Brief (MEEQ-B) for use in the current study. No psychometrics on the adapted measure were presented. Further, some aspects of the index do not appear to be a good fit (e.g., use of dated racially charged terminology [marijuana]; implication of a “high” for a product not intended to have that effect). Please include as much supporting psychometric information as possible, and significantly expand the discussion of the potential expectancy effects on observed findings.

4. Discuss regression to the mean as a threat to internal validity.

5. In the limitations, expand the discussion of potential gender effects (e.g., differential effects of cannabis based products; Sholler et al., 2020).

Sholler, D. J., Strickland, J. C., Spindle, T. R., Weerts, E. M., & Vandrey, R. (2020). Sex differences in the acute effects of oral and vaporized cannabis among healthy adults. *Addiction biology*, e12968.

8. The fact that CBD self-administration was not visually confirmed (but rather depended on incoming bottle weights and a self-report diary, both of which could be manipulated) should be included in the limitation section.

Reviewer #3 (Remarks to the Author):

This paper presents data from an open label pilot study of CBD extract for the treatment of anxiety in adults who meet clinically relevant thresholds on validated measures of anxiety. The paper is well written and is a significant contribution to the field. Upon careful review, I note the following comments and suggestions:

1) In the abstract, please include THC concentration of the product, define criteria used for "clinically significant treatment response" and provide more concrete data regarding adverse events (e.g. number/duration/severity of AEs) rather than simply stating the drug was well-tolerated with few side effects.

2) I disagree with the conclusion that "full spectrum" products have efficacy at lower doses than single extracted/synthetic compounds. Though an inverted u-shaped curve was shown in one laboratory study after acute CBD administration, no controlled study has tested doses lower than 300mg CBD in a chronic dosing paradigm as a treatment for anxiety. In the absence of data showing single chemical entity CBD products are not effective using the same dosing regimen used in this study, comparison of efficacy between single entity and full-spectrum products is not valid. I suggest you either drop or significantly alter the terminology used related to this throughout the text (abstract, summary and multiple places in the manuscript).

3) In the methods it states that participants were required to be naive to recent

cannabis/cannabinoid use, but toxicology testing was limited to the detection of THC metabolites in urine. Did you explicitly ask about or test for the use of CBD at the outset of the study? In my experience, individuals don't always endorse CBD use when asked if they use "cannabis" and it may not show up in urine tox tests targeting THCCOOH. If CBD was explicitly asked about via self-report, or if urine was also tested for CBD metabolites, this should be mentioned. If not, it should be acknowledged as a possibility that some participants could have been using CBD at baseline and during the study (but you can also argue this was unlikely given the clinical response observed).

4) Other than CBD and THC, are there any additional features of the cannabis extract that can be provided (e.g. concentrations of other minor cannabinoids or terpenoids of interest)? Please also include formulation and packaging details (e.g. what type of matrix was the extract dissolved in (e.g. sesame oil), what type of container was used, and how did participants self-administer the product).

5) At the end of the methods section, please clarify with respect to judgement of clinical significance for change scores on the BAI and OASIS. 1) Did a participant have to show a >15% change on both scales to be considered clinically improved or just one of the two? 2) Is there a precedent for the 15% threshold used here or was this arbitrarily selected for this study. If the former, please provide references to prior work using this cutoff for these assessments; if the latter, please provide a little more rationale for selection of this threshold as being indicative of clinically meaningful improvement.

6) Can you please clarify the exact timing of the Baseline assessments relative to Week 1. Were baseline assessments collected at the time medications was dispensed and exactly 1 week prior to the Week 1 assessment, or were they obtained potentially several weeks before study medication was started and the Week 1 assessment was completed?

7) The drop in anxiety scores are pretty dramatic. Though these are depicted in the figures, you may want to include the % drop in scores from Baseline to Week 1 and Week 4 in the text to underscore the magnitude of improvement noted in the study. In fact, given the magnitude of effects, consider adding the % of participants who achieved reductions of 50% or more in addition to the 15% threshold already used.

Reviewer #4 (Remarks to the Author):

This manuscript describes a single-arm open-label study of CBD for the treatment of anxiety for which a randomised controlled trial is currently underway.

I have a few comments/queries:

Text

1. Clarify to what extent the data analysis was pre-specified. Was there a statistical analysis plan in place before database lock and undertaking of analysis?
2. Please provide further detail on the power calculation, including assumptions made, in order to allow independent reproduction.
3. The BAI and OASIS are both described as primary outcome variables. Can you provide a rationale for this and explain how this is taken into account in the power calculation?
4. How was the target dose for this study identified and how does it inform the full trial?
5. Please provide further detail of the mixed effects model, specifying fixed and random effects and

including the method used for defining degrees of freedom. A limitation is the small sample size. Were diagnostic plots examined?

6. Multiple comparison correction was based on the number of time-points, but not number of outcomes. Can you clarify, are the BAI and OASIS considered co-primary?

7. The improvement in outcomes occurred within the first week of treatment. Can you comment on the potential impact of regression to the mean, particularly as high BAI and OASIS scores were inclusion criteria.

8. Where are the correlations between expectancy and outcomes? Have you considered including expectancy as a covariate in the models?

9. Was the 15% reduction in anxiety defined a priori, and what is the clinical rationale for this?

10. When describing decreases in outcomes as significant, do you mean statistically or clinically (or both) ? (Could refer to published MCID's if available)

Tables/plots

11. All tables – check number of decimal places reported is appropriate.

12. Include minimum and maximum values in Table 1

13. Figure 1 – the categories for diagnosis and severity are helpful. Suggest to include individual profiles to visualise the variation in individual trajectories. Can all outcomes be included?

14. Figure 2 - boxplots or dotplots would better illustrate the distribution in outcomes at each time-point.

Harvard Medical School
Department of Psychiatry

ASSOCIATE PROFESSOR
OF PSYCHIATRY

McLean Hospital
McLean Imaging Center

DIRECTOR
COGNITIVE AND CLINICAL
NEUROIMAGING CORE

DIRECTOR
MARIJUANA INVESTIGATIONS
FOR NEUROSCIENTIFIC DISCOVERY PROGRAM
(MIND)

STACI A. GRUBER, PH.D.

May 27, 2022

To Reviewers:

Attached please find our response and subsequent revision for the manuscript previously entitled “Clinical and Cognitive Improvement Following Treatment of Anxiety with a Full-Spectrum, High-Cannabidiol Product: Results from the Open-Label Phase of a Randomized Clinical Trial,” (COMMSMED-21-0620-T) for consideration in *Communications Medicine*. We are delighted that the Reviewers recognized the strengths of the manuscript, and appreciate the thoughtful comments generated. We hope that our revision fully addresses the Reviewers’ concerns. Individual responses to each of the Reviewer comments follow.

REVIEWER COMMENTS:

Reviewer #1:

This manuscript provides a detailed description of a well-conducted 4-week, open label pilot trial of “full-spectrum CBD” for 14 persons with elevated levels of anxiety who responded to online or social media ads to participate. Excellent assessment and outcome measures were used, and study methods resulted in excellent adherence to the study protocol with little missing data. Measures of anxiety and cognitive functioning improved substantially following 4 weeks of the study drug. Study findings are presented clearly to the reader. The findings will be of great interest to the scientific community and clinical medicine. The only primary concern with the manuscript is the overall tendency to use language and make statements that overreach what is known in the literature about CBD’s clinical effects and that overreach what can be concluded from a relatively small open label trial with study volunteers. As the authors convey in the Discussion, these findings await confirmation in a rigorously controlled study. A more conservative write up might be considered to not inadvertently mislead the readership, particularly given the political and cultural divides related to cannabis and cannabinoid compounds. Below I offer some comments and suggestions mostly related to this concern.

We thank Reviewer 1 for recognizing the strong methodology and importance of this research to the scientific community and clinical medicine. We agree with the Reviewer regarding the importance of language that does not overstate the results of this relatively small, open-label trial. Accordingly, we have adjusted the language and tone of the findings. Additionally, we have addressed specific comments and suggestions below:

1. Perhaps consider use of less superlative adjectives throughout related to promise, potential, and outcomes of CBD, e.g., the use of “great” in first line of the abstract to modify potential.

As noted above, we agree with the Reviewer, and have revised the language and tone throughout the manuscript to avoid overstating the results of this open-label trial.

- Perhaps add adherence/retention data to the abstract if there is space

We appreciate Reviewer 1's suggestion and have edited the Abstract to note high adherence/patient retention.

- For this journal, full spectrum vs. single extracted compounds, as referred to in the abstract, may need more explanation if the readership is to glean its potential importance.

We agree it is important to provide information regarding full-spectrum vs single extracted compounds. We have removed this information from the Abstract and Plain Language Summary and have expanded the Discussion section accordingly. This is also addressed in response to Reviewer 3, Question 2.

- Consider reworking the first paragraph to position CBD as a potential alternative or adjunct medication rather than suggesting that other medications are highly flawed and problematic and that CBD could replace them (i.e., a panacea).

We appreciate Reviewer 1's concern and have revised the Introduction section accordingly to both underscore the value of conventional medications for some and the potential for CBD to be an adjunctive or alternative treatment.

- p. 5 line 89. It was unclear why language switched to “medical cannabis treatment” (rather than CBD) on recreational cannabis consumers, as referred to here, aren't CBD users – this breeds unnecessary confusion for the reader (and eventually the public).

We thank Reviewer 1 for the opportunity to clarify this issue. On page 5, line 89, the term “medical cannabis treatment” was used in the text as it was directly referring to findings from observational studies of medical cannabis patients referenced in the previous sentence. Importantly, these observational studies examine general medical cannabis treatment, with medical cannabis products selected by patients themselves resulting in a variety of product types and cannabinoid profiles (i.e., not just CBD). In order to reduce confusion, this section of the manuscript has been updated with the following text:

“However, some observational studies assessing medical cannabis patients, who often report frequent use of high-CBD products, have demonstrated significant improvements on objective, clinician-administered neuropsychological assessments following initiation of treatment, particularly on measures of executive function¹⁶⁻¹⁸, suggesting medical cannabis treatment may not result in negative cognitive outcomes. However, given chronic, heavy, recreational cannabis use is associated with cognitive decrements across a range of domains, including executive functioning and memory¹⁹, understanding the impact of medical cannabis treatment (including the use of high-CBD products) on cognition is particularly important, and additional research is needed using well-validated cognitive assessments.”

- Perhaps the intro could better introduce the reader to what differentiates “whole-plant, full-spectrum, high CBD” from CBD only referred to earlier. i.e., what else is in the compound. Why is the product referred to as “novel”?

We appreciate the opportunity to provide clarity and have provided additional information regarding full-spectrum vs single extracted compounds within the Introduction as well as the Discussion section. Please see our response to Reviewer 3, Question 2.

7. P5. Line 106 refers to but does not provide a cite(s) for “observational studies” that show CBD improved EF.

We thank the Reviewer for their careful attention. Please see our response to Reviewer 1, Question 5.

8. Is it possible to list the other cannabinoids in the CBD product used in this study other than CBD and D9-THC?

We thank Reviewer 1 for this excellent point. We have added information regarding average exposure to specific cannabinoids beyond CBD and Δ 9-THC contained in the study product to Table 1. This Table was previously presented in the Supplemental section, but given its importance, we have moved it to the main manuscript.

9. Perhaps provide the rationale for considering a 15% reduction in anxiety scores “clinical improvement” – this could reflect a change from 20 to 17, which would be difficult to justify as clinically meaningful, particularly in an open label study or only 4 weeks. I do recognize that the results appear to show consistently larger reduction than 15%.

We thank Reviewer 1 for the opportunity to clarify this point. While there is no consensus on a specific threshold to define clinically significant improvement, previous research suggests that higher thresholds are often too conservative. The $\geq 15\%$ threshold was selected for the current study based on literature demonstrating that a 15% reduction in symptoms was optimal for identifying those who had responded to treatment (Johnco et al., 2015, Johnco et al., 2015). Accurately identifying response to treatment is important for clinical decision making and defining outcome in clinical trials, and these previous findings suggest that even small reductions in symptoms may indicate clinically meaningful improvement. However, as the reviewer appropriately notes, clinical improvement in the current study vastly exceeded the $\geq 15\%$ threshold. This justification has been added to the manuscript with the following text:

“Percent change relative to baseline was calculated for BAI and OASIS scores for all follow-up assessments to generate cumulative frequencies of treatment responders, defined as patients who achieved and maintained clinical improvement of $\geq 15\%$ reduction in anxiety scores on either the BAI or OASIS. This threshold was selected based on previous work demonstrating that $\geq 15\%$ symptom reduction was optimal for identifying those who had responded to treatment^{41,42}.”

10. Given the cognitive data feature prominently in the purpose of the study, perhaps the data and analyses might better be placed in the primary report, not supplemental analyses.

We strongly agree with Reviewer 1, and have moved the cognitive (and other) data from the Supplemental section to the main manuscript.

11. The issue of power raised in the methods, then again in the discussion, does not seem highly relevant to a small open label trial – particularly given all participants responded in one direction and produced highly significance results (consider omitting?). This comes across as a little defensive, unnecessarily, and adds to the trend of arguing back against basic study weaknesses that are inherent to an open label trial.

We appreciate Reviewer 1’s perspective regarding the relevance of detailed power analyses, particularly considering the large effect sizes observed and the inherent limitations of a small, open-label study. This information was requested by the journal editor to address the COVID-related change to sample size; therefore, it should likely remain within the manuscript.

Additionally, Reviewers 1 and 4 had conflicting opinions regarding the inclusion of these analyses (See Reviewer 4, Question 2). We have tried to balance all concerns within this revised manuscript and included information regarding power analyses only in the Discussion/Limitations section.

12. The discussion mentions all the limitations and weaknesses of the study, primarily related to what are always the weaknesses with an open label test of a medication. I find that the Discussion oversells the findings given these weaknesses – unnecessarily. More conservative language related to what was observed and how one might interpret the findings given the study design might be prudent. Also a more conservative discussion of the limitations without the subsequent text trying to discount the limitations would provide the reader with a more balanced presentation and interpretation of what was learned in this open label. Concluding efficacy before running and obtaining results from a rigorously controlled study is always problematic – and unfortunately in clinical science most positive findings are not supported by future trials. We can hope that will not be the case with this cannabinoid compound, but we must wait for the future results.

As suggested by Reviewer 1 above, we have revised the manuscript to remove superlative adjectives and adjusted the language to avoid overstating the results of the current study. Within the Discussion section, we have revised the text and emphasized the need for future studies, including clinical trials, to fully ascertain clinical efficacy. Noting concerns raised by all reviewers, we have revised the Discussion section to provide a more balanced perspective of limitations.

Reviewer #2:

This open label study aimed to assess the effects of a low dose full spectrum high CBD sublingual administration on anxiety and cognitive function among participants with elevated anxiety. Significant decreases in anxiety were observed at all four timepoints compared to baseline. Additionally, improvements in cognitive functioning were observed across various endpoints. These findings add to the small but growing empirical literature regarding the anxiolytic effects of CBD.

The detail in terms of methodology was appropriate for the aim of reproducibility, and the statistical approach was adequate. Below are suggestions aimed at improving the manuscript.

We thank Reviewer 2 for recognizing the strong methodology and importance of this research to the empirical literature regarding the anxiolytic effects of CBD. We have addressed specific comments and suggestions from Reviewer 2 below:

1. In the consort chart, please clarify reasons for exclusion as part of the Figure caption. Relatedly, it is unconventional to have two indices for the primary (anxiety) inclusion criterion (i.e., minimum of 16 on the Beck Anxiety Inventory [BAI] OR a minimum of 11 on the Overall Anxiety Severity and Impairment Scale [OASIS]). Provide additional detail regarding the rationale for this approach.

As requested, we have updated the consort flow chart to include the inclusion/exclusion criteria. Additionally, we have moved the flow chart from the Supplemental Section to the main manuscript. With regard to having two indices for the primary inclusion criterion, we selected both the BAI and the OASIS as these scales measure different aspects of anxiety. The BAI rates specific symptoms associated with anxiety experienced by an individual in the past week (e.g., heart racing, nervousness). The OASIS rates general anxiety (not specific symptoms) over the past week as well as the impact anxiety has had on avoidance responses, work, and social relationships. Both metrics are well-validated, standard assessment tools, and are utilized for inclusion criterion in order to consider symptom presentation and the impact symptoms have on daily functioning and quality of life.

2. Please include baseline correlations among the primary outcome variables in the supplemental material. Although the authors make a conceptual argument that these are distinct, the empirical overlap is likely quite large and may point to the need for further corrections.

We agree with Reviewer 2 regarding the importance of evaluating collinearity to ensure independence of dependent variables. In the current dataset, the primary outcome variables (BAI & OASIS) did not correlate with each other at baseline (see correlation matrix below). In addition, we examined collinearity across all anxiety variables, and noted that only BAI scores positively correlated with HAMA scores. Not surprisingly, the two STAI subscales positively correlated with each other. Given that these baseline correlations were largely non-significant, no further corrections were applied to the analyses in addition to the corrections for number of timepoints assessed.

Correlation Matrix: Baseline Anxiety	OASIS	STAI: State	STAI: Trait	HAMA
BAI	$r=.286$ $p=.322$	$r=.114$ $p=.699$	$r=-.016$ $p=.956$	$r=.694$ $p=.006$
OASIS	.	$r=.259$ $p=.372$	$r=.382$ $p=.178$	$r=.463$ $p=.095$
STAI: State	.	.	$r=.734$ $p=.003$	$r=.394$ $p=.164$
STAI: Trait	.	.	.	$r=.243$ $p=.402$

3. Expectancy effects are a potent threat to the current findings. The authors argue that expectancies measured at baseline do not relate to clinical outcomes, increasing confidence in the observed findings. That is a reasonable effort to empirically address expectancy, but it is not super compelling in the context of the current study. For example, because a CBD expectancy measure does not currently exist, the authors adapted the Marijuana Effect Expectancy Questionnaire-Brief (MEEQ-B) for use in the current study. No psychometrics on the adapted measure were presented. Further, some aspects of the index do not appear to be a good fit (e.g., use of dated racially charged terminology [marijuana]; implication of a “high” for a product not intended to have that effect). Please include as much supporting psychometric information as possible, and significantly expand the discussion of the potential expectancy effects on observed findings.

As Reviewer 2 correctly notes, a CBD-related expectancy measure does not currently exist, making it difficult to properly evaluate expectancy effects. As noted within the revised manuscript, we did not adapt the MEEQ-B, a well-validated measure that provides scores for positive and negative expectancies related to general cannabis use, but instructed patients to complete the MEEQ-B based on expectancies related to the study product. Although we recognize that the MEEQ-B was not designed to assess treatment expectancies, we selected this instrument as patients may have general and specific expectancies and biases related to cannabis use (e.g., feeling more calm, concerns about intoxication). Overall, we agree with Reviewer 2 about the critical need for well-validated expectancy measures specifically related to medical cannabis treatment; in the ongoing double-blind phase of this study, we created a measure designed to specifically assess medical cannabis treatment-related expectancies.

In order to address Reviewer 2’s concerns, we have clarified and expanded the discussion of the MEEQ-B and the necessity of properly evaluating expectancy effects, specifically those related to medical cannabis treatment. The following text has been added to the Discussion section:

“Open-label designs can be biased by treatment expectancies, as both patients and research staff are unblinded regarding study product status. Importantly, in the current trial, positive and negative expectancies at baseline did not correlate with clinical improvement for patients. However, it is important to note that an expectancy measure for medical cannabis treatment (e.g., CBD) does not exist. Therefore, we utilized the MEEQ-B, a well-validated metric designed to assess expectancies related to recreational cannabis use (i.e., THC exposure)⁴⁰, and instructed patients to rate expectancies regarding the study product instead of general cannabis/marijuana use. The MEEQ-B assesses positive (e.g., feeling calm, reducing tension) as well as negative (e.g., feeling high, altered perception) expectancies related to cannabis use, which likely (albeit indirectly) impact treatment expectancies. The MEEQ-B was selected to ensure that any bias regarding cannabis use was assessed. In order to specifically assess medical cannabis treatment-related expectancies, we created a measure currently in use in the ongoing double-blind, placebo-controlled phase of this trial.”

4. Discuss regression to the mean as a threat to internal validity.

As Reviewer 2 notes, regression to the mean can threaten internal validity. Specifically, extreme values trend toward the population mean upon repeated sampling, which can reduce confidence in the causal link between treatment and clinical outcome. However, this is a problem for any study assessing patients with clinically significant symptomatology, and can be addressed with randomized, placebo-controlled, clinical trials. In order to address Reviewer 2’s concerns, the following has been added to the Discussion section:

“It is also important to recognize that regression to the mean is potentially problematic in clinical research, particularly when assessing patients with clinically significant symptomatology, as extreme values trend toward the population mean upon repeated sampling⁴⁵. This can impact internal validity of these studies and reduce confidence in the causal link between the independent and dependent variables (e.g., treatment and clinical outcomes). Randomized, placebo-controlled, clinical trials can help differentiate between improvement related to treatment and improvement related to regression to the mean, as regression to the mean has been hypothesized to significantly contribute to placebo effects⁴⁵. The ongoing placebo-controlled phase of this trial will further address these issues. It is of note that in the current study, the large effect sizes observed for the primary clinical outcomes suggest results are not wholly attributable to regression to the mean. Further, baseline values of cognitive assessments were not extreme, suggesting that the observed improvements of executive functioning are also not likely solely based on regression to the mean.”

5. In the limitations, expand the discussion of potential gender effects (e.g., differential effects of cannabis based products; Sholler et al., 2020).

Sholler, D. J., Strickland, J. C., Spindle, T. R., Weerts, E. M., & Vandrey, R. (2020). Sex differences in the acute effects of oral and vaporized cannabis among healthy adults. *Addiction Biology*, e12968.

We agree with Reviewer 2 and have expanded the Discussion section to include additional information regarding potential sex effects with citations included. The new text is as follows:

“In the current study, patients were primarily White women with above average IQ, potentially limiting the generalizability of results. Epidemiological studies indicate that White Americans are significantly more likely to be diagnosed with general anxiety disorder⁴⁶, but evidence suggests greater persistence (≥ 12 months) of mental health disorders among non-White minorities, with lower educational attainment and birthplace (i.e., US-born) associated with greater persistence of mental health disorders⁴⁷. Additionally, lifetime prevalence statistics indicate that women are ~1.5

times more likely to have an anxiety disorder than men¹, whereas the gender distribution of our sample (78.6% female) included slightly higher numbers of women relative to population prevalence. Research on sex differences associated with CBD is limited, although evidence from preclinical and acute administration studies suggest that sex differences (and sex*age interactions) significantly impact the anxiolytic effects of THC^{48,49}. Further, cytochrome P450 (CYP) enzymes, which are responsible for many metabolic processes including drug metabolism and clearance, are significantly impacted by sex, age, and ethnicity⁵⁰, which may impact the metabolism of CBD and other cannabinoids. Future studies should confirm efficacy of CBD-containing products for anxiety in underserved and underrepresented patient samples as well as comprehensively assess potential sex-specific effects of CBD.”

6. The fact that CBD self-administration was not visually confirmed (but rather depended on incoming bottle weights and a self-report diary, both of which could be manipulated) should be included in the limitation section.

We agree with Reviewer 2 regarding the importance of proper monitoring of drug adherence. In the current trial, two methods of drug adherence were utilized and verified weekly at check-in visits: outgoing vs incoming bottle weights and self-report diaries. This methodology has been clarified in the text as follows: “Actual product use was quantified via outgoing versus incoming bottle weights cross-referenced with drug diaries, which were reviewed at weekly check-in visits.” While we acknowledge self-administration of study product has inherent limitations, these methods for monitoring drug adherence and compliance are supported by CONSORT guidelines for clinical trials. Further, using multiple methods to verify study drug adherence and compliance likely increases accuracy of data collection.

Reviewer #3:

This paper presents data from an open label pilot study of CBD extract for the treatment of anxiety in adults who meet clinically relevant thresholds on validated measures of anxiety. The paper is well written and is a significant contribution to the field. Upon careful review, I note the following comments and suggestions:

We thank Reviewer 3 for recognizing the significance of the current work. We have addressed specific comments and suggestions from Reviewer 3 below:

1. In the abstract, please include THC concentration of the product, define criteria used for "clinically significant treatment response" and provide more concrete data regarding adverse events (e.g. number/duration/severity of AEs) rather than simply stating the drug was well-tolerated with few side effects.

We appreciate Reviewer 3's comments regarding the importance of including specific details in the Abstract and have revised accordingly.

2. I disagree with the conclusion that "full spectrum" products have efficacy at lower doses than single extracted/synthetic compounds. Though an inverted u-shaped curve was shown in one laboratory study after acute CBD administration, no controlled study has tested doses lower than 300mg CBD in a chronic dosing paradigm as a treatment for anxiety. In the absence of data showing single chemical entity CBD products are not effective using the same dosing regimen used in this study, comparison of efficacy between single entity and full-spectrum products is not valid. I suggest you either drop or significantly alter the terminology used related to this throughout the text (abstract, summary and multiple places in the manuscript).

We appreciate the opportunity to clarify the importance of comparing full-spectrum to single extracted compounds given previous findings suggesting differential dose response related to product type. While a direct comparison of full-spectrum vs single extracted CBD products is beyond the scope of the open-label phase of this trial, our intention was to highlight that the significant treatment response observed in the current study occurred at a much lower dose than a previous study utilizing a CBD isolate (30mg vs 300mg). Importantly the double-blind, placebo-controlled phase of this trial includes a CBD isolate treatment arm, which will allow for a direct comparison of the full-spectrum product to a matched CBD isolate. Accordingly, we have revised the manuscript including removing statements about full-spectrum vs single extracted isolate products from the Abstract and Plain Language Summary in order to address Reviewer 3's concerns. We have also updated the Discussion section to include additional detail regarding previous studies examining full-spectrum and single extracted isolate products as well as the matched CBD isolate treatment arm in the double-blind phase of the trial (new text copied below).

“Interestingly, in the current trial, treatment response was observed at a much lower dosage (~30mg/day) than a previous trial using a single extracted CBD isolate (~300mg/day)¹². This difference may be related to the entourage effect, a term used to describe the potentially enhanced effects of cannabinoids when a variety of metabolites and closely related compounds (e.g., cannabinoids, terpenoids, flavonoids) work together synergistically¹⁵. While few studies have directly compared full-spectrum and single extracted products, research suggests that for some conditions, full-spectrum products may yield therapeutic response at lower doses and with fewer side effects. Specifically, a meta-analysis by Pamplona and colleagues¹⁴ demonstrated that patients with refractory epilepsy treated with full-spectrum, high-CBD products reported lower average dose relative to those treated with single extracted CBD isolate products. Further, single extracted CBD products were associated with more frequent reporting of mild and severe side effects relative to full-spectrum, high-CBD products. Additionally, preclinical research from Gallily and colleagues¹³ reported a bell-shaped dose-response curve for the anti-inflammatory and anti-nociceptive effects of a single extracted CBD isolate, but a linear dose-response for a full-spectrum high-CBD product (17.9% CBD, 1.1% THC, plus other cannabinoids). The primary goal of the current study was to gather safety and efficacy data on the novel, full-spectrum study product to help inform dosing for the double-blind phase; however, the double-blind, placebo-controlled phase of the trial also includes a matched CBD isolate treatment arm, which will allow for direct comparison of full-spectrum and single extracted products.”

3. In the methods it states that participants were required to be naive to recent cannabis/cannabinoid use, but toxicology testing was limited to the detection of THC metabolites in urine. Did you explicitly ask about or test for the use of CBD at the outset of the study? In my experience, individuals don't always endorse CBD use when asked if they use "cannabis" and it may not show up in urine tox tests targeting THCCOOH. If CBD was explicitly asked about via self-report, or if urine was also tested for CBD metabolites, this should be mentioned. If not, it should be acknowledged as a possibility that some participants could have been using CBD at baseline and during the study (but you can also argue this was unlikely given the clinical response observed).

We thank Reviewer 3 for the opportunity to clarify this point. As part of eligibility screening for the current trial, we administered a modified version of the Timeline Followback (TLFB), which comprehensively assesses history of cannabis and cannabinoid use. As part of this assessment, we specified that the term “cannabis” referred to marijuana, hemp, CBD, or any cannabinoid-containing product. We have updated the Methods section to include this important point with the following text: “History of previous cannabis and cannabinoid use was assessed using a modified version of the Timeline Followback²², which queried both recreational and medical cannabis use. For this assessment, we specified that the term “cannabis” referred to marijuana, hemp, CBD, or any cannabinoid-containing product.”

Overall, patients were extremely forthcoming about previous cannabis use during the TLFB, with the majority reporting some remote history of cannabis use (see Table 1); importantly, average length of abstinence was ~14 years. Additionally, while standard urine assays only assess THC-related metabolites, all patients were queried regarding their use of cannabis and cannabinoid products at each check-in visit. This has also been clarified in the text with the following:

“Additionally, patients were asked about their substance use at weekly check-in visits throughout the 4-week trial. Patients were disqualified from the trial if they endorsed use of any cannabis/cannabinoid-based products other than the study product.” One patient was disqualified from the trial for disclosing a single use of another cannabis product (see Figure 1).

We also agree with Reviewer 3 that given the large effect sizes observed in the current trial, if any patients managed to bypass the screening for baseline cannabis use, it would have been unlikely to have significantly impacted the results.

4. Other than CBD and THC, are there any additional features of the cannabis extract that can be provided (e.g. concentrations of other minor cannabinoids or terpenoids of interest)? Please also include formulation and packaging details (e.g. what type of matrix was the extract dissolved in (e.g. sesame oil), what type of container was used, and how did participants self-administer the product).

We appreciate the opportunity to provide additional details about the study product, which are now included in the manuscript (see Table 1, Methods section). For ease of reference, text is also included below.

“The study product was formulated from a full-spectrum, high-CBD base extract from cannabis provided by the National Institute on Drug Abuse. The CBD extract was homogenized into a solution containing medium chain triglyceride oil with an emulsifier, polysorbate-80, and dispensed into 30mL glass bottles. The final study product contained 9.97mg/mL CBD (1.04%) and 0.23 mg/mL THC (0.02%), confirmed by ProVerde Laboratories (Milford, MA). Information about other cannabinoids is presented in Table 1. For this 4-week trial, patients were instructed to use the calibrated dropper to self-administer 1mL of the study product sublingually and hold for a minimum of 60 seconds, 3 times per day. The targeted daily dose was 30mg CBD and <1mg THC. Actual product use was quantified via outgoing versus incoming bottle weights cross-referenced with drug diaries, which were reviewed at weekly check-in visits.”

5. At the end of the methods section, please clarify with respect to judgement of clinical significance for change scores on the BAI and OASIS. 1) Did a participant have to show a >15% change on both scales to be considered clinically improved or just one of the two? 2) Is there a precedent for the 15% threshold used here or was this arbitrarily selected for this study. If the former, please provide references to prior work using this cutoff for these assessments; if the latter, please provide a little more rationale for selection of this threshold as being indicative of clinically meaningful improvement.

We thank Reviewer 3 for the opportunity to clarify this important issue. Clinically significant improvement was defined as $\geq 15\%$ reduction of symptoms on either the BAI or OASIS. For more information regarding the precedent for selecting the 15% threshold for clinical improvement, please see our response to Reviewer 1, Question 9.

6. Can you please clarify the exact timing of the Baseline assessments relative to Week 1. Were baseline assessments collected at the time medications was dispensed and exactly 1 week prior to the Week 1 assessment, or were they obtained potentially several weeks before study medication was started and the Week 1 assessment was completed?

We thank Reviewer 3 for this important question. The baseline data presented in the manuscript were all collected at the same visit when study drug was dispensed; these data were not collected weeks before the study intervention was initiated.

7. The drop in anxiety scores are pretty dramatic. Though these are depicted in the figures, you may want to include the % drop in scores from Baseline to Week 1 and Week 4 in the text to underscore the magnitude of improvement noted in the study. In fact, given the magnitude of effects, consider adding the % of participants who achieved reductions of 50% or more in addition to the 15% threshold already used.

We agree with Reviewer 3 that adding the percent change from Baseline to Week 4 underscores the magnitude of improvement noted in the study. These percent change values have been added to Tables 3 and 4 and Supplemental Table 2. Unfortunately, given space constraints we could not also include percentage data for patients who achieved $\geq 50\%$ reductions of anxiety symptoms.

Reviewer #4:

This manuscript describes a single-arm open-label study of CBD for the treatment of anxiety for which a randomised controlled trial is currently underway. I have a few comments/queries:

We thank Reviewer 4 for their careful consideration of the current work. We have addressed specific comments and suggestions from Reviewer 4 below:

1. Clarify to what extent the data analysis was pre-specified. Was there a statistical analysis plan in place before database lock and undertaking of analysis?

We appreciate the opportunity to clarify this important point. The current trial was approved and monitored by the Mass General Brigham Institutional Review Board, which requires a statistical analyses plan as part of the approval process for the study protocol before research can begin.

2. Please provide further detail on the power calculation, including assumptions made, in order to allow independent reproduction.

We agree with the Reviewer about the importance of providing information to allow for independent reproduction of power analyses and have revised the manuscript accordingly (see text below). Importantly however, Reviewer 1 and Reviewer 4 had conflicting opinions regarding the inclusion of these analyses (See Reviewer 1, Question 11); we have therefore attempted to balance the concerns of each reviewer in this revision.

“Additionally, restrictions on in-person research due to the onset of the COVID-19 pandemic resulted in enrollment of a slightly smaller sample size than suggested by our a priori power analyses. However, power analyses for 5 repeated-measurements (at power=.90 and $\alpha=.05$) indicated that shifting the sample size from 16 to 14 patient completers only slightly impacted the required effect size ($\eta^2=.11$ vs $\eta^2=.12$). Further, the lowest observed effect size for the primary assessments of anxiety in the current analyses was $\eta^2=.38$ (for STAI state anxiety), more than double the required effect size from both power analyses. This suggests the study is well-powered

to assess the primary outcome variables despite the slightly smaller final sample size than originally anticipated.”

3. The BAI and OASIS are both described as primary outcome variables. Can you provide a rationale for this and explain how this is taken into account in the power calculation?

We thank Reviewer 4 for this important question. The rationale for using both the BAI and OASIS has been addressed in a previous question (see Reviewer 2, Question 1). In regard to the power analyses, the original power analyses were based on standard alpha level (.05) without corrections. However, the large effect sizes observed in this trial indicate sufficient statistical power in the current analyses.

4. How was the target dose for this study identified and how does it inform the full trial?

We thank the Reviewer for this important question regarding the target dose for this clinical trial. This dose was based on clinicians’ observations among a medical cannabis patient collective using a similar sublingual tincture for various physical and psychological symptoms, including anxiety. Within the collective, over 100,000 doses were provided; no adverse events were reported, and patients noted anecdotal improvements in anxiety and sleep. Importantly, a primary goal of the open-label phase of our study was to confirm efficacy and tolerability of the full-spectrum study product at 10mg/ml, t.i.d., which is supported by the current data.

5. Please provide further detail of the mixed effects model, specifying fixed and random effects and including the method used for defining degrees of freedom. A limitation is the small sample size. Were diagnostic plots examined?

We thank Reviewer 4 for the opportunity to clarify this important point. For the current trial, linear mixed model analyses (SPSS syntax: MIXED) with first-order autoregressive AR(1) covariance structures (reference group=baseline) were used for the analyses. The “mixed model” name is a bit of a misnomer for the current analyses as the repeated-measures effect of visit was the only factor included in the model. The MIXED procedure was selected for the current analyses in order to be consistent with planned analyses for the double-blind, placebo-controlled phase of the trial, which will include treatment arm as a random effect in the models in addition to repeated-measures effect of visit. We agree with Reviewer 4 that the small sample size is a limitation of the current trial; this is mentioned within the Limitations section. However, the large effect sizes observed in this trial indicate sufficient statistical power in the current analyses. Additionally, the observed effect sizes and variance in the study did not suggest a need to plot the residuals; therefore, diagnostic plots were not examined in the current study.

6. Multiple comparison correction was based on the number of time-points, but not number of outcomes. Can you clarify, are the BAI and OASIS considered co-primary?

We thank Reviewer 4 for this important question. The rationale for using a correction based on number of timepoints, but not number of visits ($p \leq 0.1$) was addressed in a previous question (see Reviewer 2, Question 2). Both the BAI and OASIS are considered primary outcome measures in the current study (see Reviewer 2, Question 1).

7. The improvement in outcomes occurred within the first week of treatment. Can you comment on the potential impact of regression to the mean, particularly as high BAI and OASIS scores were inclusion criteria.

We agree with Reviewer 4 that regression to the mean can threaten internal validity. A discussion about the impact of regression to the mean has been provided in a previous question (see Reviewer 2, Question 4).

8. Where are the correlations between expectancy and outcomes? Have you considered including expectancy as a covariate in the models?

We agree with Reviewer 4 about the importance of including the MEEQ-B expectancy correlations in the manuscript; these correlations are now included in Supplemental Table 1. It is important to note that none of these expectancy correlations were significant. However, if significant relationships had been detected between expectancy effects and clinical outcome, we would have included expectancy as a covariate in the models.

9. Was the 15% reduction in anxiety defined a priori, and what is the clinical rationale for this?

We thank Reviewer 4 for the opportunity to clarify this important issue. Please see our response to Reviewer 1, Question 9 for additional information regarding the rationale for selecting this a priori threshold.

10. When describing decreases in outcomes as significant, do you mean statistically or clinically (or both) ? (Could refer to published MCID's if available)

We thank Reviewer 4 for the opportunity to clarify this important issue. Overall, when discussing significance, we are referring to statistical significance. In the manuscript, clinical significance is only discussed in terms of treatment response with the threshold of $\geq 15\%$ reduction of symptoms. We have edited the manuscript to ensure that "clinical significance" is noted in these cases. Importantly, as discussed previously (see response to Reviewer 1, Question 9), the 15% threshold for clinical significance was based on literature demonstrating that maximum agreement with treatment response criteria was achieved at $\sim 15\%$ reduction of symptoms.

11. All tables – check number of decimal places reported is appropriate.

We appreciate the Reviewer's attention to detail, and have checked and revised the decimal places throughout the manuscript. The standard 2 decimal places are used for all data except for frequency data, in which case 1 decimal place is standard. Given the large effect sizes observed in the trial, we have used 3 decimal places when reporting significance values.

12. Include minimum and maximum values in Table 1

We have updated Table 1 to include minimum and maximum values.

13. Figure 1 – the categories for diagnosis and severity are helpful. Suggest to include individual profiles to visualise the variation in individual trajectories. Can all outcomes be included?

We thank Reviewer 4 for recognizing the strengths of Figure 1. Unfortunately, given space limitations, individual profiles could not be included in the manuscript. However, we have included additional data from the analyses of clinical scales in Table 3 and Supplemental Table 2.

14. Figure 2 - boxplots or dotplots would better illustrate the distribution in outcomes at each time-point.

Given Reviewer 4's concerns about the line graphs in Figure 2, we have removed this figure from the revised version of the manuscript. These data are now presented in Table 3.

We thank the editors and the reviewers for their time and consideration of our manuscript. If you require any more information, please do not hesitate to contact me.

Sincerely,

Staci A. Gruber

REVIEWERS' COMMENTS:

Reviewer #1 (Remarks to the Author):

The authors provided a comprehensive and adequate response to all my previous queries and concerns, and modified the manuscript accordingly. I have no further comments.

Reviewer #2 (Remarks to the Author):

The authors were responsive to reviewer concerns. I have no additional suggestions at this time.

Reviewer #3 (Remarks to the Author):

The revision has addressed all my prior concerns

Reviewer #4 (Remarks to the Author):

Thank you for addressing my queries, I have no further comments.